# Image information optimization processing based on fractional order differentiation and WT algorithm

Qiong Long *

School of General Education, Chengdu Jincheng College, Chengdu, China

* longqiong2022@163.com

## Abstract

As one of the most important ways for humans to perceive the world, images contain a wealth of visual information. Digital image processing is a technology that uses computer methods to process and enhance photographs in order to extract meaningful information and improve image quality. However, current image processing techniques have poor performance in processing complex images. To improve the quality of complex images, research proposes an image information optimization processing method based on fractional order differentiation and WT algorithm. Image edge detection and image fusion are important technologies in the field of image processing, with wide application value. Therefore, the study is based on wavelet transform algorithm and fractional order differentiation to perform edge detection and image fusion. The results revealed that when the study used the four evaluation metrics of information entropy, recall, mean square error, and precision to evaluate the effectiveness of image edge detection, the Sobel operator had the highest precision of detection recall, and the smallest information entropy and mean square error. The method achieved an 80% recall rate, a minimum information entropy of 3.13, a highest detection precision of 78.9%, and a minimum mean square error of 152. The average gradient, information entropy, spatial frequency, mutual information of the method adopted by the study for image fusion was compared with other methods in case of different groups of images. The method adopted by the study for image fusion provided the best results. The precision of the proposed method edge detection by the study was higher and the performance of image fusion was better and effective in improving the quality of the image.

## 1. Introduction

As the visual basis of human perception of the world, images play a crucial role in the information-rich society of the 21st century by enabling people to express, receive, and transmit information [1–2]. However, people can perceive the world more

**Data availability statement:** All relevant data are within the article and its supporting information files.

**Funding:** The author(s) received no specific funding for this work.

**Competing interests:** The authors have declared that no competing interests exist.

**List of Abbreviations:** FRWT: Fractional Wavelet Transform; WT: Wavelet Transform; FOD: Fractional order derivative; FO: Fractional order; SMC: Sliding mode controller; LF: Low frequency; TF: Temporal frequency; DWT: Discrete wavelet transform; RL: Riemann-Liouville; G-L: Grimwald-Letnikov; ED: Edge detection; RO: Roberts Operators; PO: Prewitt Operators; SO: Sobel Operators; LOG-O: LOG Operators; CA: Canny algorithm; GF: Gaussian filtering; GAV: Gradient amplitude value; PP: Pixel point; CO: Canny Operators; HF: High frequency; CWT: Continuous wavelet transform; OS: Original signal; MV: Mean value; var: variance; cov: Covariance; SA: Sobel algorithm; RA: Roberts algorithm; LOG-A: LOG algorithm; PA: Prewitt algorithm; LFW: Labeled Faces in the Wild.

accurately and objectively by using digital image processing technologies. It is possible to restore clarity and brightness to blurred or even invisible photographs using image enhancement techniques. There are two important steps in image processing, namely image edge detection and image fusion [3]. Many scholars have pointed out that current image processing techniques may distort or lose details when processing complex images. Moreover, the noise and complex background of images make it difficult for current image processing techniques to accurately segment and recognize targets [4–5]. The use of wavelet transform multi-scale processing technology for image denoising and feature extraction has demonstrated efficient performance. With the increase of data volume, how to effectively remove noise and interference from images while maintaining image details remains a challenge. When using fractional differentiation for edge detection, not only can it effectively extract the edge information of the image, but it can also preserve weak edge information in smooth regions. The image fusion method based on fractional wavelet transform (FRWT) can remove noise and interference in the image while maintaining image details, improving the quality and clarity of the image [6]. The image quality requirements in fields such as medical diagnosis and geological exploration are relatively high. There are problems in multi-focus image fusion, such as easy detail loss and artifacts at image edges. In view of this, in order to improve the quality of multi focus image fusion, avoid the loss of local details and edge artifacts, a method based on wavelet transform (WT) algorithm and fractional order differentiation (FOD) is proposed to optimize the information of medical and satellite images.

When studying the optimization processing of image information based on fractional order differentiation and WT algorithms, the innovation of this study lies in the combination of fractional order differentiation and WT algorithm. Compared with traditional integer order differentiation, fractional order differentiation can more flexibly adjust the order of differentiation, thereby finely controlling the local feature extraction of images. The main contribution of this study is to provide a high-performance solution for edge detection and fusion of digital images, which greatly promotes the development of digital image processing research and the intelligence of image fusion. It provides an effective method for the application of image edge detection in displacement monitoring equipment, medical images and remote sensing images.

The research will investigate the image information optimization processing method based on FOD and WT algorithm from four aspects. First, an overview of the state of FOD and WT algorithm research is presented. Second is the research on image information optimization processing methods for FOD and WT algorithms. Next comes the experimental verification of the suggested research design. Lastly, there is a summary of the research.

## 2. Related works

Fractional calculus generalizes differential and integral operations by unifying them into a single fractional-order (FO) derivative of arbitrary order, and is commonly used in finance, engineering, and science [7]. To address mismatched disturbances in DC/DC buck converters, X. Lin et al. presented a FO sliding mode management approach based on

a higher-order nonlinear disturbance observer. The findings showed that the technique suggested by the study enhanced the system's transient performance while lessening the system's sensitivity to mismatch disturbances [8]. B. Long et al. presented a fuzzy FO non-singular terminal sliding mode controller (SMC) for lower control limit-grid connected converter (LCL-GCC) grid current management in an effort to solve the issue that chattering in SMC tended to excite low-frequency (LF) unmodeled dynamics and may deteriorate tracking precision. The study was based on state estimation of minimally sampled sensors by Kalman filtering. The results indicated that the proposed controller of the study could converge quickly with high tracking precision and robustness [9]. A novel finite-time FO command filtering implementation approach was proposed by X. You et al. [10]. The stabilizing function's FO derivatives did not need to be analytically calculated in the command filtering implementation technique. The outcomes showed that the suggested strategy could guarantee that the tracking error converges in a finite amount of time to a tiny neighborhood around the origin [11]. B. Babes et al. suggested an adaptive fuzzy FO non-singular terminal SMC for synthetic robustness in an effort to enhance a DC-DC buck converter's output voltage tracking control performance. The study combined the fractional calculus with the hybrid control method of non-singular terminal sliding mode controller (NTSMC). The study's suggested method enhanced the conventional SMCs' resilience to disturbances and parameter fluctuations [12]. To prevent repeated resonances that could impair system stability and power quality, M. A. Azghandi et al. presented a delay based lossless FO virtual capacitor analysis and design. The capacitor used as a resonance damper for a multiple parallel paralleling current-source inverters (CSI) system. The suggested capacitor, according to the data, offered more degrees of freedom to improve the control's robustness and frequency behavior [13].

For time-frequency (TF) analysis and signal processing, the WT is a perfect instrument since it offers a frequency-varying "TF" window [14]. A WT real-time defect detection technique based on machine learning was presented by Y. Ma et al. for naval DC pulsating loads. The study's suggested method could locate further flaws that could cause anomalous disruptions in the load current profile [15]. To create four distinct types of insulation failures in power cables, M. H. Wang et al. suggested a convolutional neural network power cable fault detection method based on discrete wavelet transform (DWT) and chaotic system. The findings showed that the suggested approach was capable of accurately identifying fault status changes in power cables and fault signal variations in power cables in real time [16]. R. P. Medeiros et al. found some delays in phase-based differential protection during fault occurrence due to phase convergence and proposed a new time-domain power transformer differential protection based on Clarke and WT that could be used for any power transformer. The suggested approach was effective, quick, easy to use, and independent of the differential current's fundamental and harmonic components, according to the results [17]. Empirical WT and differential fault energy were used by J. Gao and colleagues to decompose the differential fault energy using empirical WT, which allowed for the distinction between high impedance faults and normal disturbances in distribution systems. The findings showed that the study's suggested method could accurately identify high impedance defects from normal disturbances [18]. Xu et al. proposed an interference suppression technique for vehicle-mounted millimeter-wave radar in the adjustable Q-factor WT domain. The technique involved optimizing the model using a split-enhanced Lagrangian contraction algorithm, which addressed the issue of increased noise level caused by the distortion of radar echoes in the presence of mutual interference. As evidenced by the data, the suggested strategy lessens radar echo interference [19].

In summary, the above literature demonstrates the powerful application potential and advantages of fractional calculus and WT techniques in multiple fields. These studies have achieved significant results in improving system performance, enhancing robustness, and solving specific problems. However, there are still some challenges and limitations in terms of implementation complexity, computational cost, model dependency, dynamic response performance, long-term stability, and adaptability to complex environments, which need to be addressed and optimized in further research in the future. The proposed method combines fractional order differentiation and wavelet transform algorithms, which can effectively extract image edge information while preserving weak edge information in smooth areas. This method introduces fractional order differentiation, allowing the algorithm to adjust the differentiation order more flexibly and thus more finely control the extraction of local features in the image.

## 3. Image information optimization processing

### 3.1. Image edge detection based on fod

There are many ways of defining fractional calculus and the commonly used forms of definition are Riemann-Liouville (RL) definition, Grümwald-Letnikov (G-L) definition, and Caputo definition [20–21]. The Grumwald-Letnikov fractional calculus definition is obtained by higher-order differential generalization, which is a fundamental concept in fractional calculus, providing an extension of the traditional notion of integer-order derivatives. The G-L formula is important in fractional calculus because it provides a way to compute non-integer order derivatives. With FO derivatives, the derivative's order might be a real number as opposed to only an integer [22]. The G-L formula defines the FO derivative $D^a f(x)$ of the function $f(x)$ at the point $x$. The FO derivative $D^a f(x)$ is illustrated in Equation (1).

$$D^a f(x) = \lim_{h \to 0} \sum_{k=0}^{\infty} (-1)^k \binom{a}{k} f(-kh + x)$$

(1)

In Equation (1), $\binom{a}{k}$ denotes a binomial coefficient and $a$ denotes a real number. $h$ denotes the integration time step. $k$ denotes a non-negative integer. When $k$ is equal to 0, the binomial coefficient is defined as 1. The RL integral is a FO promotion of the successive indefinite integrals [23]. Equation (2) expresses the RL integral of the function $f(x)$.

$$I^\alpha f(x) = \frac{1}{\Gamma(\alpha)} \int_c^x f(t) (x-t)^{\alpha-1} dt$$

(2)

In Equation (2), $\Gamma$ denotes the gamma function and $x$ denotes the integral variable. t denotes the integration time, $c$ denotes the initial point of integration taken. $\alpha$ is the derivatives of the original function and $I^\alpha$ is the bounded linear operator. Caputo fractional derivative is one of the generalizations of integer order derivatives, which actually means any order derivative greater than or equal to 0 [24]. Caputo fractional derivative is shown in Equation (3).

$$d_{Caputo}^v = \frac{1}{\Gamma(n-v)} \int_c^x \frac{f^{(n)}(\tau)}{(t-\tau)^{v-n+1}} d\tau$$

(3)

In Equation (3), $n$ denotes the smallest integer greater than $v$. $0 \leq n-1 < v < n$, $f^{(n)}(\tau)$ is the $n$ order derivative of $f(x)$. In image processing, difference of Gaussian (DOG) is a very useful technique for detecting edges and corner points in an image. The difference between these two results is obtained by convolving the image with Gaussian kernels of different standard deviations [25]. DOG calculation is shown in Equation (4).

$$G(x, y) = -\frac{y^2 + x^2}{2\sigma^2} + \frac{1}{\sqrt{2\pi\sigma^2}} e$$

(4)

In Equation (4), $\sigma$ denotes the variance and $(x, y)$ denotes the coordinates. The smaller $\sigma$ is the higher the peak of the Gaussian function. The larger $\sigma$ is, the smoother the peaks are and the blurrier the image convolution effect is. There is a large change in the gray scale (GS) of the image edge, and the gradient value can reflect the drastic degree of GS change, so it is necessary to find the gradient in each direction of the image, and thus determine the edge region of the image. Currently the commonly used operators for edge detection (ED) and extraction are Roberts operator (RO), Prewitt operator (PO), Sobel operator (SO), LOG operator (LOG-O) and Canny algorithm (CA). The fractional gradient operator can finely control the local feature extraction of signals or images by adjusting the differential order, thereby achieving optimal performance in different application scenarios. Compared with integer order differentiation, fractional gradient operators can better capture subtle changes in signals, especially when dealing with complex nonlinear systems. Secondly, the fractional gradient operator also has memory, that is, it not only considers the value of the current point, but also considers the values of all past

points. In addition, the fractional gradient operator can more effectively suppress noise while extracting signal features. This is because the fractional gradient operator can balance the smoothness of the signal and the accuracy of feature extraction by adjusting the order, thereby preserving important features of the signal while removing noise. The following is the specific ED process. First, the acquired image is transformed into a GS image, and then noise is removed from the image using Gaussian filtering (GF). Subsequently, an appropriate ED method is chosen to identify the image, and the edges that are identified are binaryized to produce an image with black and white edges. Finally, the detected edges are connected and detailed to output the processed edge image. The steps of image ED are shown in Fig 1.

The RO is a gradient algorithm that is based on cross-differentials and is also referred to as the cross-differential algorithm [26]. The approach is often used to process steep, low-noise images and discovers edge lines through the computation of local differences. When the image edges are near plus or minus 45 degrees, the algorithm performs better. When RO performs ED on an image, first the selected operator template is used to operate on the image to be detected and calculate the gradient magnitude value (GAV) of the image. Secondly, a suitable threshold is selected and the GAV of the image is judged point by point. Points that meet the criteria are assigned a value of 1, while the remaining points are assigned a value of 0. Finally, the edge image of the image to be detected is obtained. The bias derivation of RO for $x$, $y$ direction is shown in Equation (5).

$$\begin{cases} G_x = -f(x, y) + f(x + 1, y + 1) \\ G_y = -f(x, y + 1) + f(x + 1, y) \end{cases}$$

(5)

The LOG-O smooths the image by GF function to suppress the effect of noise [27]. The bias derivation of the LOG-O for the $x$, $y$ directions is shown in Equation (6).

$$\nabla^2 G(x, y) = \left[ \frac{x^2 + y^2 - 2\sigma^2}{\sigma^4} \right] e^{\frac{x^2 + y^2}{2\sigma^2}}$$

(6)

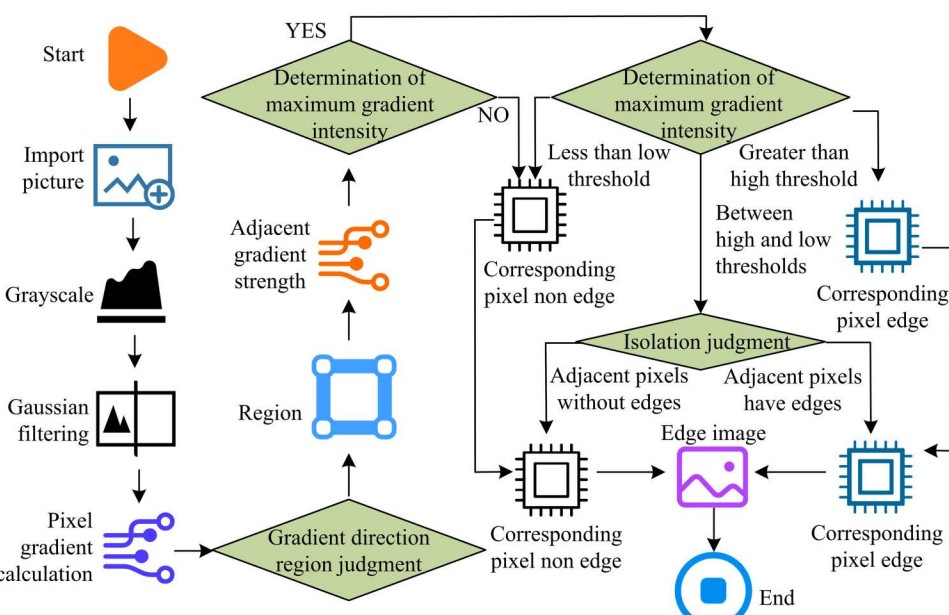

**Fig 1. Improved image edge detection algorithm flow based on fractional gradient operator.**

In Equation (6), $\sigma$ denotes the variance and $e^{(\cdot)}$ denotes the exponential function. Fig 2 displays the LOG-O's ED step.

PO is used to achieve ED by using the difference generated by the gray values of the pixels in a particular region. The image is first subjected to a convolution operation in order to extract the gradient values of each pixel point (PP) in both the horizontal and vertical directions. Secondly, the binarized edge image is obtained by thresholding the gradient intensity for segmentation [28]. The ED effects of the PO are more noticeable in the horizontal and vertical dimensions since it employs a $3 \times 3$ template for the pixel values in the region. It works well for identifying photographs with a higher level of noise and GS gradient. The bias derivation of PO for $x$, $y$ direction is shown in Equation (7).

$$\begin{cases} G_x = f(x+1,y) + f(x+1,y-1) - f(x-1,y) \\ -f(x-1,y+1) - f(x-1,y-1) + f(x+1,y-1) \\ G_y = f(x,y+1) + f(x-1,y+1) - f(x,y-1) \\ -f(x-1,y-1) - f(x+1,y-1) + f(x+1,y+1) \end{cases} \tag{7}$$

The goal of CA is to find an optimal ED solution or to find the location in an image where the gray intensity varies the most [29]. Three factors are mostly used to determine optimal ED: lowest response, high localizability, and low error rate. The specific steps of ED by Canny operator (CO) are as follows. First, the image is smoothed and its gradient is found using GF. Second, double thresholding is employed to determine the possible boundaries and non-maximum suppression technique is utilized to filter out the non-edge pixels. Finally, hysteresis technique is utilized to track the boundaries. Fig 3 depicts the CO ED process.

Combining differential derivation and GF, the SO is a discrete differential operator [30]. By labeling certain points in the region that exceed a certain number as edges, depending on how bright or dark they are adjacent to the edge of the image, the SO is used to approximate the brightness and darkness of an image. The SO adds the concept of weight to the PO and considers that the proximity of neighboring points has a different effect on the current PP. The image is sharpened and the edge contours are emphasized when the PPs are closer together, since they have a greater effect on the current pixel. The weighted difference of the gray levels of a PP's upper and lower, left and right neighbors reaches an extreme value near the edge, which is how SO determines the edge. The expression of FOD of SO is shown in Equation (8).

$$G(x,y) = \left[ \frac{\partial^v f(x,y)}{\partial x^v} - \frac{\partial^v f(x,y)}{\partial y^v} + \right. \\ \left. a\frac{\partial^v f(x+1,y+1)}{\partial x^v} - a\frac{\partial^v f(x+1,y+1)}{\partial y^v} \right] \tag{8}$$

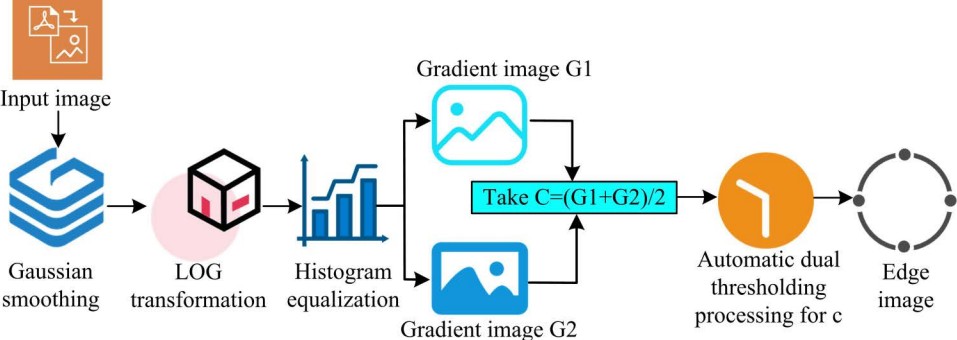

**Fig 2. LOG operator edge detection steps.**

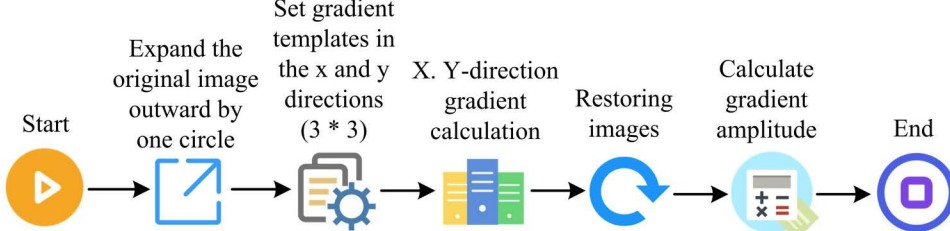

**Fig 3. Canny operator edge detection steps.**

In Equation (8), $a = 2\sqrt{2} + 1$. $v$ indicates that the derivative with respect to $x, y$ is of order $v$. The specific steps for ED of SO are as follows. The original image is first subjected to Gaussian blurring before being transformed into a GS image. Second, the Sobel function is used to get the derivatives in both the x and y axes. The derivatives in the x and y directions are finally overlaid. The rate of change of the image's GS is reflected by spatial frequency (SF), and a larger SF suggests a clearer image and higher-quality fused image. The FO SF is shown in Equation (9).

$$FSF = \sqrt{(FRF)^2 + (FCF)^2}$$

(9)

In Equation (9), $FSF$ denotes FO SF. $FRF$ denotes FO row frequency and $FCF$ denotes FO column frequency. To accurately refine the wide edges of the gradient amplitude image, it is necessary to perform non-maximum suppression on the gradient image and remove some non-edge points. The direction of the gradient is discriminated into four directions: 1, 2, 3, and 4, and replace them with directions. Map the gradient direction of the pixel to the nearest direction. Map the gradient direction of the pixel to the nearest direction. If the gradient direction of the central pixel belongs to the second direction and is greater than the average gray value and the maximum gray value of the gradient image, then the pixel is determined to be a local maximum point. In addition, to determine the appropriate threshold, the Otsu algorithm is used to calculate the threshold. This method traverses the gray values of the entire image and selects the gray value corresponding to the maximum inter-class variance as the threshold. The formula for calculating the inter-class variance is shown in Equation (10).

$$\begin{cases} \sigma = p_0 (u_0 - u) + p_1 (u_1 - u) \\ u = p_0 * u_0 + p_1 * u_1 \end{cases}$$

(10)

In Equation (10), $\sigma$ represents inter class variance. $p_0$ represents the probability that the grayscale is less than the segmentation threshold. $u_0$ represents the average grayscale of images with grayscale less than the segmentation threshold. $u_1$ represents the average grayscale of images with grayscale greater than the segmentation threshold. $p_1$ represents the probability that the grayscale is greater than the segmentation threshold. $u$ represents the total average grayscale of the image.

### 3.2. FRWT-based image fusion

The study of image ED based on FOD followed by IF based on FRWT is able to accurately extract edge information. WT is a brand-new transform analysis technique that is perfect for processing and analyzing signal TF. While FRWT is a multi-resolution analysis of signal TF domain. Performing FRWT on a signal is equivalent to passing the signal through an FO filter bank consisting of a scale function and wavelet functions with different scales to obtain the LF portion of the signal and the high-frequency (HF) portion of the signal in different frequency bands [31–32]. Its primary capabilities include the ability to fully highlight the features of specific problem aspects through transformation and the ability to localize time

and SF analysis. The method eventually accomplishes time subdivision at HF and frequency subdivision at LF, which can automatically adjust to the requirements of TF signal analysis, thereby focusing on arbitrary features of the signal. The signal is gradually refined at numerous scales by telescopic translation operations. Compared with the traditional WT, it has better resolution, stronger localization and better anti-noise performance, especially suitable for processing signals with complex spatio-temporal features. The image processing based on FRWT is shown in Fig 4.

In WT theory, DWT and continuous wavelet transform (CWT) are the two main transformations [33]. The signal is transformed in the discrete time or space domain by DWT, whereas the signal is transformed in the continuous time or space domain by CWT. The discretized WT performs and is more stable in real time [34–35]. The basic principle of DWT is to discretize the CWT, which can decompose the signal or image into components of different frequencies [36]. DWT has the ability to analyze both time and frequency domains. Compared with general pyramid decomposition, DWT image decomposition can obtain LF information in horizontal, vertical and diagonal directions while extracting LF information of the image. By choosing the mother wavelet reasonably, DWT can make DWT more effective in extracting significant information such as texture and edge while compressing noise. Pyramid decomposition has information correlation between scales, while DWT has higher independence in different scales. For IF based on DWT, firstly, WT is performed on the source image. Secondly, the transformed coefficients are merged according to certain rules. Ultimately, the fused image is obtained by applying the wavelet inverse transform to the joined coefficients. In Fig 5, the IF based on DWT is displayed.

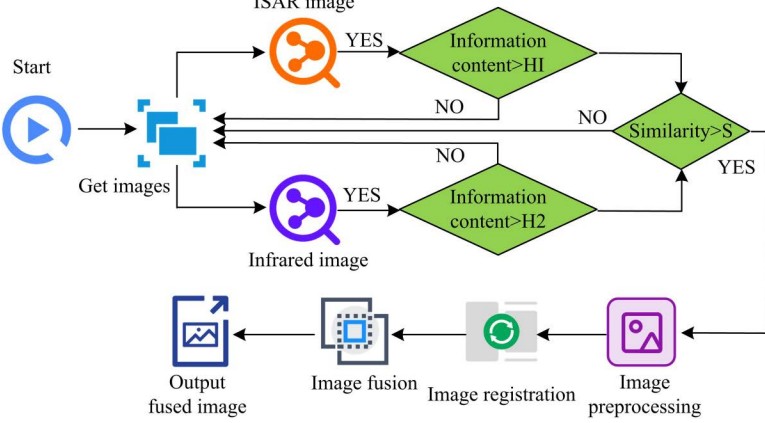

**Fig 4. Image based on fractional wavelet transform.**

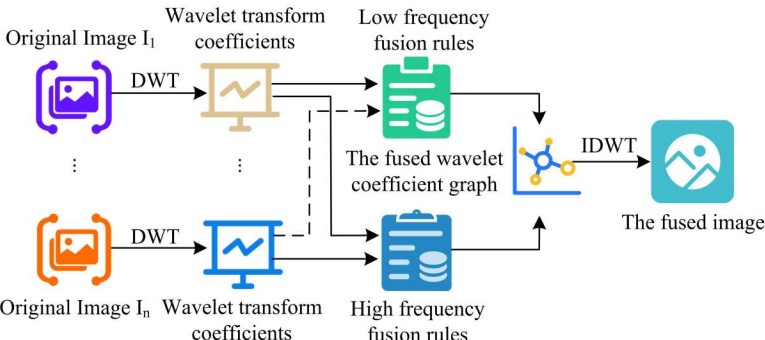

**Fig 5. Image fusion based on discrete wavelet transform.**

 

WT is a representation of the original signal (OS) using a scale function. The scale grows larger and larger as the scale level drops, representing the OS in an increasingly erratic and hazy manner while also increasing the gap between the OS. It is used as a scale function to meet the requirements that the scale function is orthogonal to its integer translation, and that the function space grown at low scales is nested within high scales. The scale function expression is shown in .

$$\varphi(x) = \sum_{k \in Z} h_\varphi(k) \sqrt{2} \varphi(2x - k)$$

(11)

In Equation (11), $k \in Z$, $k$ determines the position of $\varphi(x)$. $\varphi(x)$ denotes the scale function and $h_\varphi(k)$ denotes the expansion coefficient. The relationship between scale space and wavelet space is shown in Equation (12).

$$\psi(x) = \sum_k h_\psi(k) \sqrt{2} \varphi(2x - k)$$

(12)

In Equation (12), $\psi(x)$ denotes the wavelet function and $h_\psi(k)$ denotes the wavelet function coefficients. In the discrete case, the approximation coefficients are calculated in Equation (13).

$$T_\varphi(j, k) = \sum_n h_\varphi(n - 2k) T_\varphi(j + 1, n)$$

(13)

In Equation (13), $T_\varphi(j, k)$ represents the approximation coefficient. $j \in Z$, $j$ determines the width and height, and $n = 1, 2, \ldots$. The fine coefficients are calculated in Equation (14).

$$T_\psi(j, k) = \sum_n h_\psi(n - 2k) T_\psi(j + 1, n)$$

(14)

In Equation (14), $T_\psi(j, k)$ denotes the fine coefficient. Bootstrap filtering is an image filtering technique that can achieve edge smoothing with double filtering by filtering the initial image with one bootstrap image. The bootstrap filtering uses one bootstrap image to generate weights, which are processed on the input image. This process expression is shown in Equation (15).

$$q_i = \sum_j W_{ij}(I) \cdot p_j$$

(15)

In Equation (15), $q$ is the output image. $I$ is the bootstrap image. $P$ is the input image. $i$ and $j$ are the index of the PP in the image. $W$ denotes the weights. The bootstrap filtering implementation steps are shown in .

As shown in , firstly, the boxFilter filter is used to complete the correlation coefficient parameter (CCP). The mean value (MV) includes autocorrelation MV, cross-correlation MV, bootstrap image MV, and the original image MV that needs to be filtered. CCP calculates based on MV, including autocorrelation variance (var) and cross-correlation covariance (cov). Secondly, calculate the coefficients of the window linear transformation parameters a and b. The MV of parameters a and b is calculated according to the formula. Finally, use these parameters to obtain the output image of the bootstrap filter. The regions of the image where the brightness or gray value varies slowly are represented by the LF component. This expansive, level region of the picture characterizes the majority of the image and provides a thorough gauge of the overall intensity of the image. Equation (16) illustrates the fusion of the LF coefficient.

$$LL = w_1 * LL_1 + w_2 * LL_2$$

(16)

In Equation (16), $w_1$, $w_2$ denote the weighting coefficients. $LL_1$, $LL_2$ denote the LF coefficients. $LL$ is the fused LF coefficient. The areas of the image that fluctuate significantly, such as the borders, noise, and detail, are corresponding to the HF components. HF coefficient fusion first calculates the absolute value of the HF coefficients, and second selects the HF coefficients with the largest absolute value at each position as the fusion result. The specific steps of FRWT based IF are

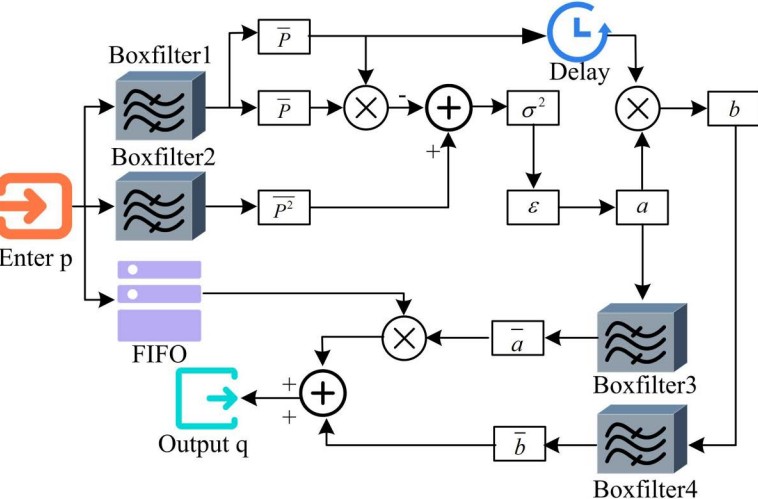

**Fig 6. Steps for implementing guided filtering.**

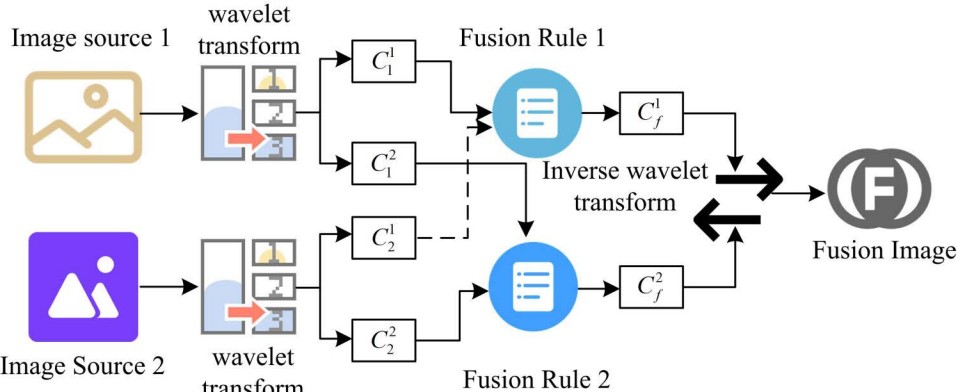

**Fig 7. Image fusion based on fractional wavelet transform.**

as follows. Firstly, the input images are pre-processed, such as gray-scaling and normalization. Then DFRWT decomposition is performed on the two input images to get the LF coefficients and LF coefficients of the images. Secondly, the LF coefficients are weighted and fused. The weights can be determined according to the image quality, information content and other factors. For example, for better quality images, higher weights can be assigned. Maximum value fusion is performed for LF coefficients. That is, the HF coefficients with the largest absolute value at each position are selected as the fusion result. Finally, the fused LF coefficients and HF coefficients are inverted by DFRWT to obtain the fused image. The specific steps of FRWT-based IF are shown in Fig 7.

## 4. Quality analysis of image edge detection and fusion

### 4.1. Experimental parameter setting

The study employs Sobel algorithm (SA), Roberts algorithm (RA), LOG algorithm (LOG-A), Prewitt algorithm (PA) and CA for three sets of image ED simulation experiments. Considering the diversity and representativeness of the dataset,

as well as the comprehensiveness of performance evaluation, the selected datasets for the experiment include Labeled Faces in the Wild (LFW) dataset, Kaggle PCB defect dataset, and Rice image dataset. Compared to the datasets used in other literature, experiments conducted on the aforementioned three datasets can verify the applicability of the algorithm in various practical applications. For example, edge detection of facial images can be applied to facial recognition and expression analysis; edge detection of circuit board images can be used for industrial quality control; edge detection of rice images can be utilized for image analysis in the agricultural field. To test the generalization performance of the proposed algorithm, the LFW dataset and Kaggle PCB Defects dataset will be used for training, while the Rice Image dataset will be used as an independent dataset to test the algorithm's performance. The LFW dataset is used to test the edge detection performance of the proposed method, the Kaggle PCB dataset is used to verify the edge detection effect of the research algorithm on images with rich edges, and the Rice Image dataset is used to verify the algorithm's noise resistance performance. Among them, Experiment 1 performs ED on face images. Experiment 2 performs ED on complex circuit board images. Experiment 3 performs ED on rice grain images with 0.05 pretzel noise added. Fifteen sets of images of size 512*512 are selected for fusion in the study. The three methods NSCT method, PNCC method and L_P method are also selected for comparison and analysis with the FRWT method used in the study. The operating system chosen for the experiment is Windows 11X64, RAM is 16GB, CPU is 12th Gen Intel(R) Core(TM) i7-12700H @ 2.30GHz, the graphics card model is NVIDIA GeForce RTX 3060, and the graphics card memory is 16G, and the software chosen is MATLAB R2018a. FO P takes the values of 0.4 and 0.5, respectively.

### 4.2. Quality analysis of image edge detection based on fod

The study employs RA, SA, PA, LOG-A and CA to perform ED on three images with different levels of complexity. Among them, the image structure of Experiment 1 is more complex, the image of Experiment 2 has more complex textures at the edges, and the image structure of Experiment 3 is simpler. B is the eligible edge points and A is the total detected points. The results of ED are shown in Table 1. The B/A value of SA is the smallest in all the three sets of experiments and ED has the highest precision rate. When the detected image structure is more complex, the B/A value of SA is the smallest and the B/A value of PA is the highest, which are 0.0143, 0.0915 ($p<0.05$), respectively. When the image edges have more complex textures, the B/A value of SA is the smallest and the B/A value of RA is the highest, which are 0.0219, 0.3039 ($p<0.05$), respectively. When the detected image structure is simple, SA has the smallest B/A value and PA has the highest B/A value, which is 0.0146, 0.0270 ($p<0.05$), respectively.

The execution efficiencies of RA, SA, PA, LOG-A and CA are shown in Table 2 when the complexity of the detected images is different. When the structure of the detected image is more complex, the RA has the lowest execution efficiency and the CA has the highest execution efficiency of 0.0497 and 0.0518 ($p<0.05$), respectively. When the image edges have more complex textures, the PA is executed with the lowest efficiency and the CA is executed with the highest efficiency with 0.0598 and 0.0801 ($p<0.05$), respectively. When the structure of the detected image is relatively simple, the PA is executed with the lowest efficiency and the CA is executed with the highest efficiency with 0.0363 and 0.0492 ($p<0.05$), respectively. Overall, the performance of the algorithm proposed in the study on independent datasets is not

**Table 1. Quantitative analysis results of edge detection.**

| Image Name | B/A of PA | B/A of RA | B/A of LOG-A | B/A of CA | B/A of SA |
|---|---|---|---|---|---|
| Experiment 1 | 0.0915 | 0.0880 | 0.0652 | 0.0248* | 0.0143* |
| Experiment 2 | 0.2934* | 0.3039* | 0.0868* | 0.0270* | 0.0219* |
| Experiment 3 | 0.0270* | 0.0222* | 0.0224* | 0.0152* | 0.0146* |

Note: * indicates that $p<0.05$.

**Table 2. Comparison of algorithm execution efficiency.**

| Image Name | PA | RA | LOG-A | CA | SA |
|---|---|---|---|---|---|
| Experiment 1 | 0.0505 | 0.0497 | 0.0513* | 0.0518* | 0.0500 |
| Experiment 2 | 0.0598* | 0.0614* | 0.0698* | 0.0801* | 0.0739* |
| Experiment 3 | 0.0363* | 0.0379* | 0.0489* | 0.0492* | 0.0472* |

Note: * indicates that $p < 0.05$.

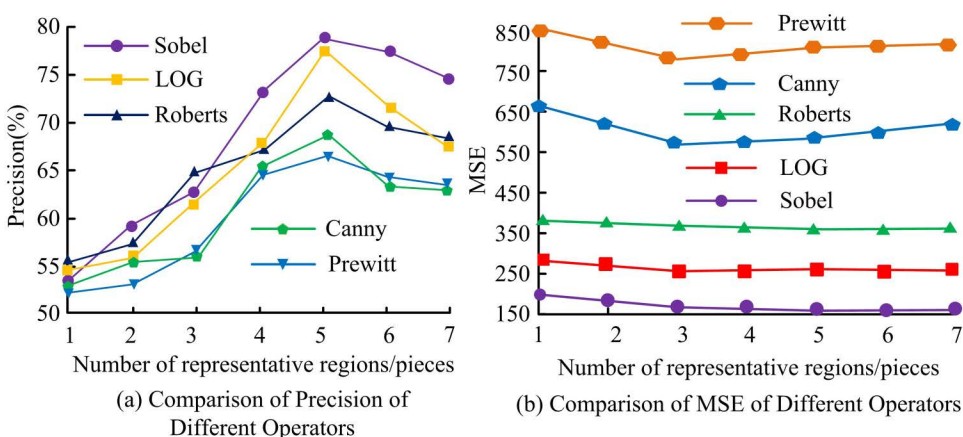

(a) Comparison of Precision of Different Operators

(b) Comparison of MSE of Different Operators

**Fig 8. Image edge detection MSE with accuracy for different operators.**

significantly different from that of the LFW dataset and Kaggle PCB Defects dataset, indicating that the model has good generalization performance.

The study adopts two evaluation metrics, mean square error (MSE) and precision rate to evaluate the effect of image ED. In this, the study detects different representative regions of added noise image. The precision rate denotes the probability that the boundary pixels generated in ED are the real boundary pixels and its statistical results are shown in Fig 8. In Fig 8(a), with the increase of representative regions, the precision rate of image ED using various techniques exhibits an increasing and subsequently declining trend. When 1–5 regions are selected as representative regions, the precision rate of image ED by different methods shows a gradual increasing trend. Compared with other methods, SO has the highest detection precision rate and PO has the lowest detection precision rate. The highest detection precision rate is 78.9% for SO and 66.3% ($p < 0.05$) for PO. In Fig 8(b), the MSE curves of different operators for image ED fluctuate little as the representative region increases. SA has the smallest MSE and PA has the largest MSE, with SA's MSE range being [775,850] and PA's MSE range being [152,200] ($p < 0.05$).

The study adopts two evaluation metrics, information entropy and recall, to evaluate the image ED effect. Among them, the recall rate indicates the probability of detecting a real boundary pixel over all real boundary pixels. The statistical results are shown in Fig 9. In Fig 9(a), the recall of different methods for image ED increases with the increase of representative regions. When 5–7 regions are selected as representative regions, the fluctuation of the recall curve of different methods for image ED is small. Compared with other methods, SO has the highest detection recall and RO has the lowest detection recall. The highest detection recall is 80% for SO and 67.9% ($p < 0.05$) for PO. In Fig 9(b), the information entropy curves of different operators for image ED fluctuate less as the representative region increases. SA has the minimum information entropy and PA has the maximum information entropy. The minimum information entropy of SA and

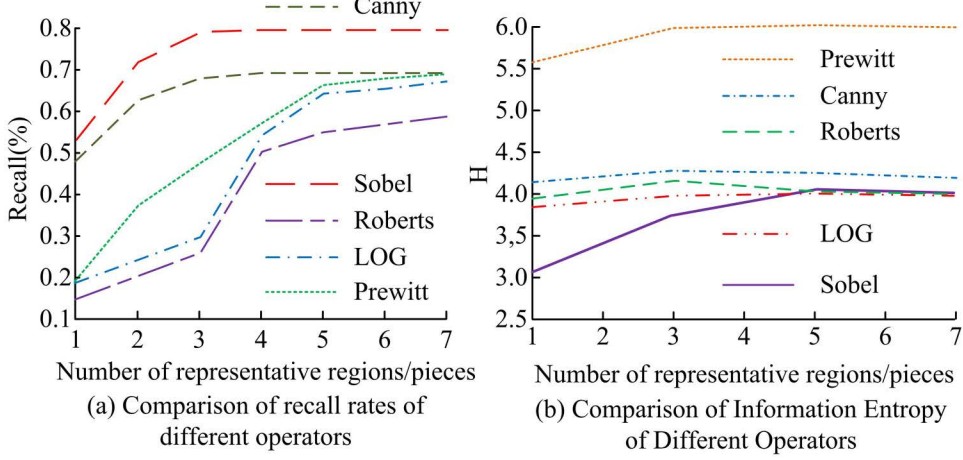

Fig 9. Information entropy and recall of image edge detection by different operators.

Table 3. B/A of different edge detection algorithms.

| Image Name | TBSWT-FM | ICO | Our Method |
|---|---|---|---|
| Experiment 1 | 0.036* | 0.041* | 0.015* |
| Experiment 2 | 0.042* | 0.039* | 0.022* |
| Experiment 3 | 0.033* | 0.028* | 0.015* |

Note: * indicates that $P < 0.05$.

PA is 3.13 and 5.61 ($p < 0.05$), respectively. The maximum information entropy of SA and PA is 3.99 and 5.98 ($p < 0.05$), respectively.

To further analyze the performance of the proposed edge detection algorithm, it is compared with the Triple B-spline wavelet transform and Franklin moment (TBSWT-FM) and the improved canny operator (ICO). The test results are shown in Table 3.

According to Table 3, compared to other algorithms, the proposed edge detection algorithm has a smaller B/A ratio, which does not exceed 0.022 ($p < 0.05$). It can be concluded that the proposed edge detection algorithm in this study has higher accuracy and higher edge integrity. The effects of edge detection for the different datasets are shown in Figure Fig 10.

As shown in Fig 10, compared to other algorithms, the proposed method not only achieves accurate edge detection of ordinary images, but also improves the edge detection effect of images with rich edges. In addition, this method has good resistance to salt and pepper noise, and its noise profile is significantly reduced during the detection process.

## 4.3. FRWT-based quality analysis of image fusion

To verify the performance of the algorithm proposed in the study, five typical methods are selected for comparative analysis with the proposed method. Three sets of multi focus images with a size of 512*512 are selected for fusion in the experiment, with fractional order p values of 0.4 and 0.5. The quantitative evaluation indicators based on WT image fusion are compared with existing methods, and the statistical results are shown in Table 4. Among them, MI represents the mutual information between two random variables, SF represents SF, and QAB/F represents the degree of image edge preservation. Observing Table 4, it can be seen that when the fractional order p = 0.4, the MI, QAB/F, SF, and values of the WT based image fusion method are the best, which are 5.0183, 0.6981, and 12.6585, respectively. The image fusion methods

based on L_P have the lowest values of MI, QAB/F, SF, and are 4.6235, 0.5817, and 7.1638, respectively. When the fractional order p = 0.5, the MI, QAB/F, SF, and values of the WT based image fusion method are optimal, with values of 5.1205, 0.7405, and 9.0975, respectively. The image fusion methods based on L_P have the lowest values of MI, QAB/F, SF, and are 4.2372, 0.4987, and 5.1348, respectively.

The average gradient, information entropy of the method used in the study for IF is compared with other methods in case of different groups of images. In Fig 10, the statistical data are displayed. In Fig 11(a), the average gradient of the method adopted by the study for IF is higher than the other methods in the case of different image groups. The higher the average gradient, the more image layers and the clearer the image. FRWT, PNCC, NSCT and L_P methods have the highest average gradient of 3.75, 3.19, 3.28 and 2.91 ($p < 0.05$), respectively. In Fig 11(b), the information entropy of the method used in the study for IF is lower than the other methods in the case of different image groups. The lesser the

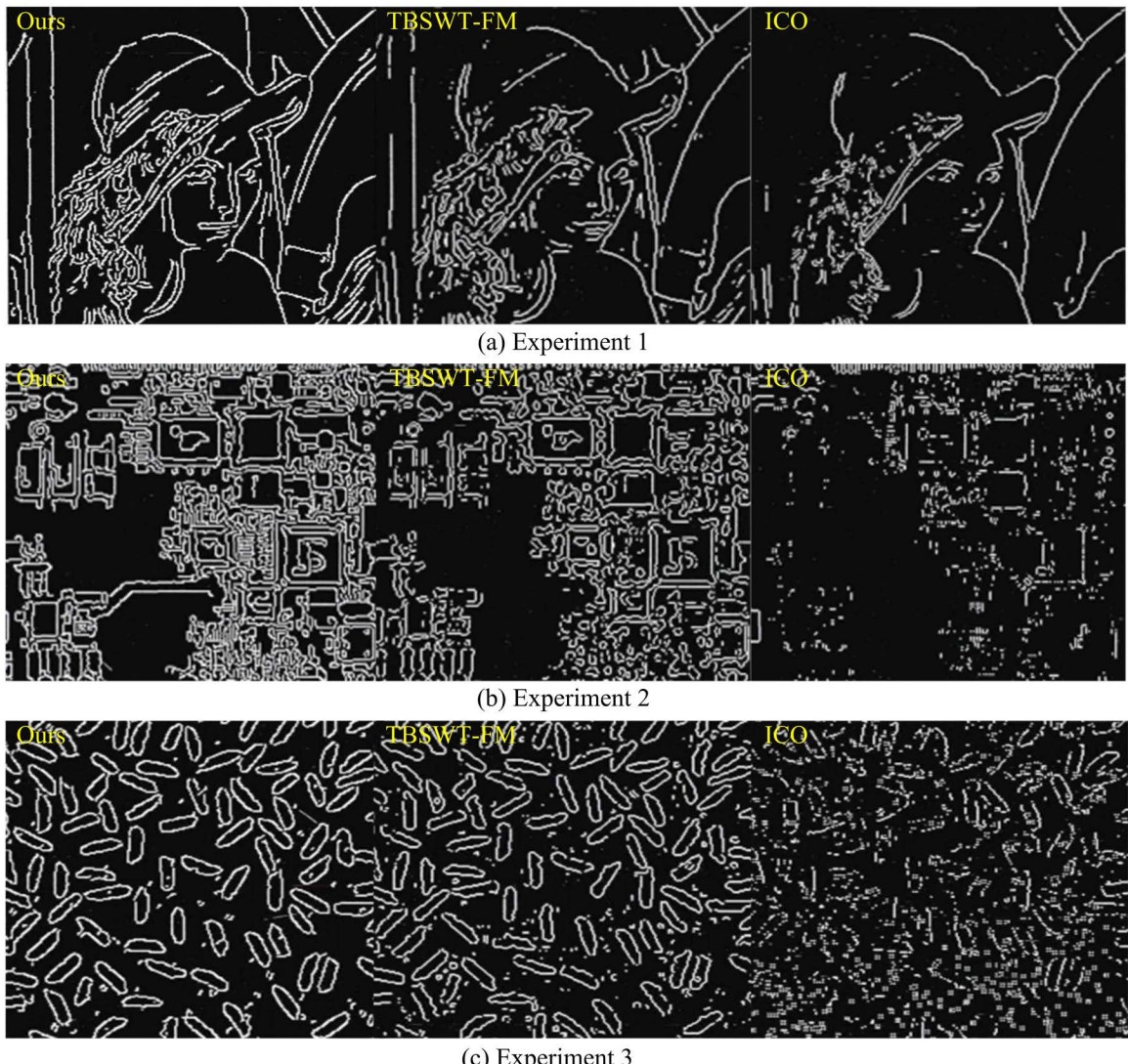

(a) Experiment 1

(b) Experiment 2

(c) Experiment 3

**Fig 10. Edge detection effects for different datasets.**

**Table 4. The image fusion quality of six algorithm.**

| Fusion method | P = 0.4 | | | P = 0.5 | | | Reference |
|---|---|---|---|---|---|---|---|
| | MI | QAB/F | SF | MI | QAB/F | SF | |
| FRWT | 5.018 | 0.698* | 12.659* | 5.121* | 0.741* | 9.098* | / |
| NSCT | 4.923 | 0.681 | 9.033* | 4.814* | 0.675* | 7.976* | Lawrance N et al [37] |
| WT | 4.823 | 0.611* | 8.620* | 4.743* | 0.654* | 7.688* | Pramanik S. et al [31] |
| DWT | 4.767 | 0.602* | 7.598* | 4.533* | 0.520* | 5.983* | R. Ahmadian et al [33] |
| PNCC | 4.823 | 0.611* | 8.620* | 4.743* | 0.654* | 7.688* | KP B et al [38] |
| L_P | 4.767 | 0.602* | 7.598* | 4.533* | 0.520* | 5.983* | Song M et al [39] |

Note: * indicates that $p < 0.05$.

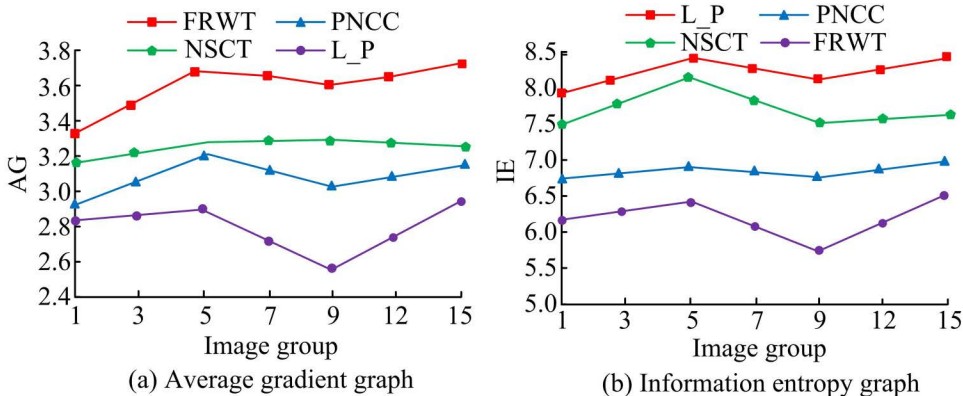

(a) Average gradient graph

(b) Information entropy graph

**Fig 11. Average gradient and information entropy for different image fusion methods.**

information entropy, the lesser the IF. FRWT, PNCC, NSCT and L_P methods have the least information entropy of 5.61, 6.59, 7.50 and 8.10 ($p < 0.05$), respectively.

In the case of different groups of images, the study used the method for IF of SF, mutual information is compared with other methods. In Fig 11, the statistical data are displayed. In Fig 12(a), the space of the method adopted by the study for IF is higher than the other methods in the case of different image groups. The image's GS's rate of change is reflected in the SF. Greater clarity and improved fused image quality are indicated by the larger SF. The highest SF is 6.820, 6.7836.805 and 6.748 ($p < 0.05$) for FRWT, PNCC, NSCT and L_P methods, respectively. In Fig 12(b), the mutual information of the methods used in the study for IF is lower than the other methods in case of different image groups. FRWT, PNCC, NSCT and L_P methods have minimum mutual information of 0.630.78, 0.94 and 1.14 ($p < 0.05$), respectively.

The bootstrap filtering determines the size of the filtering window. With different filter radii, the study used the metric SF to evaluate the effect of fusion of different five sets of images. The statistical results are shown in Fig 13. SF decreases with increasing filter radius. When the filter radius is greater than 6, the SF converges gradually. When the filter radius is 2, the SF values of the first, second, third, fourth and fifth group of images are 6.417, 6.376, 6.369, 6.310, and 6.275 respectively. When the filter radius is 6, the SF values of the first, second, third, fourth and fifth group of images are 6.404, 6.367, 6.335, 6.296, and 6.234 respectively. When the filter radius is 10, the SF values of the first, two, three, four and five groups of images have SF values of 6.378, 6.336, 6.324, 6.273, and 6.206, respectively.

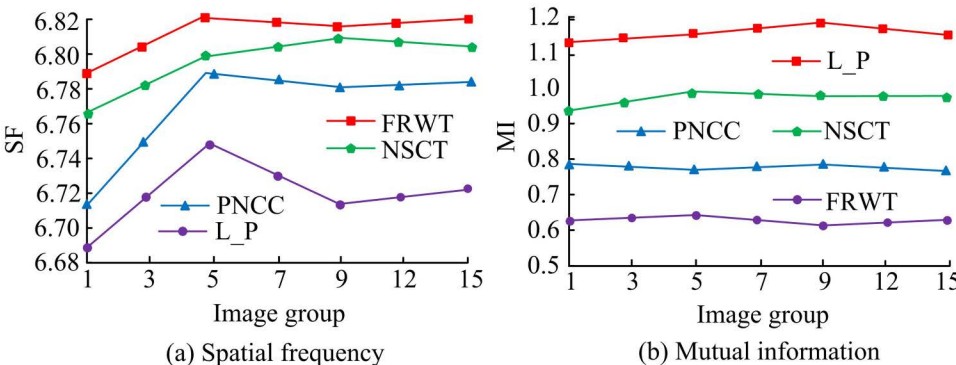

**Fig 12. The spatial frequency and mutual information of the different image fusion methods.**

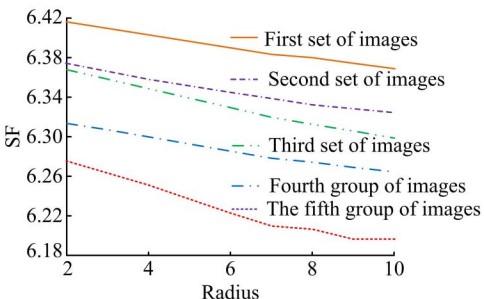

**Fig 13. The impact of different radii on fusion results.**

**Discussion.** The study proposed a method for optimizing image information processing that is based on FOD and WT algorithms. This method has demonstrated effectiveness in the domains of image edge detection and image fusion. Through experimental verification, this method performs well in image edge detection and image fusion, especially in key indicators such as detection accuracy, information entropy, recall rate, and mean square error (MSE), which are superior to traditional methods. X. Luo et al. proposed a fusion method based on quaternion WT and feature level copula model for image fusion problems. This method was developed to avoid the visual quality degradation of images caused by false information, as compared with other methods [40]. D. Agrawal et al. proposed an image fusion method based on DWT. This approach was developed to improve the visibility and quantitative performance of fused images, as compared with other methods [41]. However, despite significant achievements, there are still some potential optimization directions and application extensions worth further exploration. To further improve the performance of this method and apply it to a wider range of fields, such as video processing and real-time applications. In real-time applications, such as surveillance systems and autonomous driving, the real-time and accuracy of image processing are crucial. To achieve real-time processing, the methods proposed in the research can be simplified and optimized. For example, reducing the number of layers in wavelet decomposition or using fast fractional order differentiation algorithms. In addition, the combination of hardware acceleration technology can further improve real-time processing capabilities. Meanwhile, to adapt to the constantly changing environment in real-time applications, online learning mechanisms can be introduced to enable algorithms to dynamically adjust parameters to adapt to new image features. Different image information processing optimization methods are shown in Fig 14.

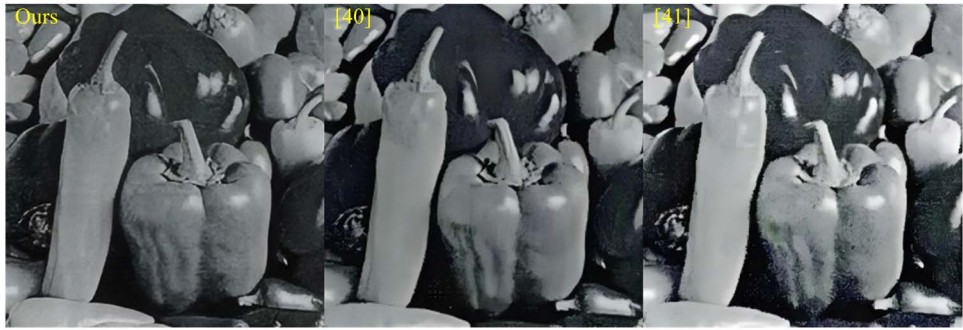

**Fig 14. Different optimization methods for image information processing.**

As can be seen from Fig 14, compared with literature [40] and literature [41], the image information optimization quality of the proposed method is better, which not only retains the new solution intact, but also basically has no noise interference.

## Conclusion

To facilitate image analysis, image storage, image transmission and improve image quality, the study proposed an image information optimization processing method based on FOD and WT algorithm. The study used FOD based approach for detecting the edges of the image. Among them, RO, SO, PO, LOG-O, and CA were used to extract the edges of the image. Second, FRWT based method was used for fusion of images. The results revealed that when the study used the four evaluation metrics of information entropy, recall rate, MSE, and precision rate to evaluate the effectiveness of image ED, the SO had the highest precision rate for detection recall rate, and the lowest information entropy and MSE. The method achieved an 80% recall rate, a minimum information entropy of 3.13, a highest detection precision rate of 78.9%, and a minimum MSE of 152. The WT based IF methods had the best MI, QAB/F, and SF values when the FO took the values of 0.4 and 0.5. When the structure of the detected image was complex, RA had the lowest execution efficiency and CA had the highest execution efficiency of 0.0497 and 0.0518, respectively. PA, RA, and SA had execution efficiencies of 0.0505, 0.0513, and 0.0500, respectively. Compared with other methods, the precision rate and execution efficiency of ED of the research proposed method are improved, and the IF is more effective. Due to the fact that only 2D and grayscale images were considered when constructing the proposed algorithm, its ability to handle higher dimensional or more complex image data is not yet clear. In view of this, it is considered to introduce directional fractional wavelet transform in the future to improve the algorithm's ability to process high-dimensional images.

## Author contributions

**Data curation:** Qiong Long.

**Software:** Qiong Long.

**Writing – review & editing:** Qiong Long.

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
