## [Decision Letter · Decision Letter 0]

23 Jan 2025

PONE-D-24-54977Image Information Optimization Processing Based on Fractional Order Differentiation and WT AlgorithmPLOS ONE

Dear Dr. Long,

Thank you for submitting your manuscript to PLOS ONE. After careful consideration, we feel that it has merit but does not fully meet PLOS ONE’s publication criteria as it currently stands. Therefore, we invite you to submit a revised version of the manuscript that addresses the points raised during the review process.

The manuscript has been evaluated by two reviewers, and their comments are available below. The reviewers have raised a number of major concerns. In particular, they request improvements to the writing, further discussion and the addition of statistical significance testing.

Could you please carefully revise the manuscript to address all comments raised?

We look forward to receiving your revised manuscript.

Kind regards,

Helen Howard

Staff Editor

PLOS ONE

Journal Requirements:

Reviewers' comments:

Reviewer's Responses to Questions

**Comments to the Author**

1. Is the manuscript technically sound, and do the data support the conclusions?

Reviewer #1: Yes

Reviewer #2: Yes

2. Has the statistical analysis been performed appropriately and rigorously? 

Reviewer #1: Yes

Reviewer #2: Yes

3. Have the authors made all data underlying the findings in their manuscript fully available?

Reviewer #1: Yes

Reviewer #2: Yes

4. Is the manuscript presented in an intelligible fashion and written in standard English?

Reviewer #1: Yes

Reviewer #2: Yes

5. Review Comments to the Author

Reviewer #1: Paper is nicely written

All experimental analysis is done nicely

Paper is of good quality. Literature is done in detailed.

Methodology is explained in details. Results shown in graphical as well as tabular form.

Reviewer #2: The proposed method, combining fractional order differentiation (FOD) and wavelet transform (WT), is innovative and technically sound. The integration of edge detection and image fusion techniques is well-justified and aligns with current challenges in complex image processing.

However, further discussion on the method's generalizability to other datasets and scenarios would enhance its impact.

The results are adequately supported by relevant metrics such as precision, recall, and information entropy. While these provide a strong foundation, adding statistical significance testing (e.g., t-tests or ANOVA) would improve the rigour of your analysis.

While the manuscript is mostly clear, specific phrases and sentences are awkwardly structured or repetitive. For instance:

"The highest recall rate was 80%, the minimum information entropy was 3.13..." could be simplified to "The method achieved an 80% recall rate and a minimum information entropy of 3.13."

Avoid redundancy, such as "the study proposed method edge detection by the study."

Ensure consistency in technical terminology, and avoid using acronyms without proper explanation.

The figures and tables effectively support the narrative but could benefit from more explicit captions that summarize their findings.

Ensure all axes and labels are legible, particularly in performance comparison graphs (e.g., Figures 8–11).

The discussion section should expand on how the proposed method could be optimized or applied to other domains like video processing or real-time applications.

Address potential limitations of the method, such as computational efficiency or dependency on parameter tuning.

6. PLOS authors have the option to publish the peer review history of their article (what does this mean? ). If published, this will include your full peer review and any attached files.

**Do you want your identity to be public for this peer review?** For information about this choice, including consent withdrawal, please see our Privacy Policy .

Reviewer #1: No

Reviewer #2: No

---

## [Author Response · Author response to Decision Letter 1]

7 Mar 2025

The manuscript has been modified.

---

## [Decision Letter · Decision Letter 1]

25 Mar 2025

PONE-D-24-54977R1Image Information Optimization Processing Based on Fractional Order Differentiation and WT AlgorithmPLOS ONE

Dear Dr. Long,

Thank you for submitting your manuscript to PLOS ONE. After careful consideration, we feel that it has merit but does not fully meet PLOS ONE’s publication criteria as it currently stands. Therefore, we invite you to submit a revised version of the manuscript that addresses the points raised during the review process.

Please revise the paper based on reviewer comments. Also consider the editor comments below.

1. A more detailed literature on the study should be added at the end of the Related works section. After this process, the difference of this study from the literature should be interpreted more clearly.

2. Although there are many different methods that can be used for image edge detection in the literature, why a method based on feature fractional gradient operator was used in this study and its originality should be stated more clearly.

3. The results obtained from the scope of the study and the metric types seem to be at an appropriate level when compared to the literature.

4. The steps for implementing guided filtering are clearly stated and clearly reveal the quality of the study.

5. The reason for the dataset preferences used in the study and their places in the literature compared to other datasets in the literature should be stated more clearly.

6. The hardware or software (toolbox etc.) used in the study should be stated more clearly and detailed.

We look forward to receiving your revised manuscript.

Kind regards,

Fatih Uysal, Ph.D.

Academic Editor

PLOS ONE

Additional Editor Comments (if provided):

Please revise the paper based on reviewer comments. Also consider the editor comments below.

1. A more detailed literature on the study should be added at the end of the Related works section. After this process, the difference of this study from the literature should be interpreted more clearly.

2. Although there are many different methods that can be used for image edge detection in the literature, why a method based on feature fractional gradient operator was used in this study and its originality should be stated more clearly.

3. The results obtained from the scope of the study and the metric types seem to be at an appropriate level when compared to the literature.

4. The steps for implementing guided filtering are clearly stated and clearly reveal the quality of the study.

5. The reason for the dataset preferences used in the study and their places in the literature compared to other datasets in the literature should be stated more clearly.

6. The hardware or software (toolbox etc.) used in the study should be stated more clearly and detailed.

Reviewers' comments:

Reviewer's Responses to Questions

**Comments to the Author**

1. If the authors have adequately addressed your comments raised in a previous round of review and you feel that this manuscript is now acceptable for publication, you may indicate that here to bypass the “Comments to the Author” section, enter your conflict of interest statement in the “Confidential to Editor” section, and submit your "Accept" recommendation.

Reviewer #3: (No Response)

Reviewer #4: (No Response)

2. Is the manuscript technically sound, and do the data support the conclusions?

Reviewer #3: (No Response)

Reviewer #4: Partly

3. Has the statistical analysis been performed appropriately and rigorously? 

Reviewer #3: I Don't Know

Reviewer #4: N/A

4. Have the authors made all data underlying the findings in their manuscript fully available?

Reviewer #3: (No Response)

Reviewer #4: Yes

5. Is the manuscript presented in an intelligible fashion and written in standard English?

Reviewer #3: Yes

Reviewer #4: Yes

6. Review Comments to the Author

Reviewer #3: The authors have clearly stated the proposed method.

The findings obtained demonstrate the effectiveness of the study.

Up-to-date and sufficient references have been added.

The authors have responded constructively to the comments of other reviewers.

Reviewer #4: The referee comments for the study titled "Image Information Optimization Processing Based on Fractional Order Differentiation and WT Algorithm" are as follows.

*The purpose of the study should be clearly explained.

*The effect of the proposed method should be shown on sample images.

*The discussion section should be made more understandable with images and tables.

7. PLOS authors have the option to publish the peer review history of their article (what does this mean? ). If published, this will include your full peer review and any attached files.

**Do you want your identity to be public for this peer review?** For information about this choice, including consent withdrawal, please see our Privacy Policy .

Reviewer #3: No

Reviewer #4: No

---

## [Author Response · Author response to Decision Letter 2]

23 Apr 2025

The manuscript has been modified.

---

## [Editor Report · Decision Letter 2]

25 Apr 2025

Image Information Optimization Processing Based on Fractional Order Differentiation and WT Algorithm

PONE-D-24-54977R2

Dear Dr. Long,

We’re pleased to inform you that your manuscript has been judged scientifically suitable for publication and will be formally accepted for publication once it meets all outstanding technical requirements.

Kind regards,

Fatih Uysal, Ph.D.

Academic Editor

PLOS ONE

Additional Editor Comments (optional):

Considering the authors’ detailed responses to the editor and reviewer feedback, the improvements made during revision, and the overall quality of the manuscript, we have decided to accept the paper for its strong potential to contribute to the field and the robustness of its final version.
---

## [Editor Report · Acceptance letter]

PONE-D-24-54977R2

PLOS ONE

Dear Dr. Long,

I'm pleased to inform you that your manuscript has been deemed suitable for publication in PLOS ONE. Congratulations! Your manuscript is now being handed over to our production team.

Kind regards,

on behalf of

Dr. Fatih Uysal

Academic Editor

PLOS ONE